Exact integer linear programming solvers outperform simulated annealing for solving conservation planning problems

http://orcid.org/0000-0003-3191-7869 Schuster Richard 1 2 richard.schuster@glel.carleton.ca
http://orcid.org/0000-0002-4716-6134 Hanson Jeffrey O. 3
http://orcid.org/0000-0001-8929-7776 Strimas-Mackey Matthew 4
Bennett Joseph R. 1
1 Department of Biology, Carleton University , Ottawa, ON , Canada
2 Ecosystem Science and Management Program, University of Northern British Columbia , Prince George, BC , Canada
3 School of Biological Sciences, University of Queensland , Brisbane, QLD , Australia
4 Cornell Lab of Ornithology, Cornell University , Ithaca, NY , USA
Boyer Alison
Electronic publication date: 2020 May 27
Publication date: 2020
Volume: 8
Electronic Location ID: e9258
Received 2020 Jan 16; Accepted 2020 May 8
Copyright: © 2020 Schuster et al.
Copyright year: 2020
Copyright holder: Schuster et al.
License: This is an open access article distributed under the terms of the Creative Commons Attribution License, which permits unrestricted use, distribution, reproduction and adaptation in any medium and for any purpose provided that it is properly attributed. For attribution, the original author(s), title, publication source (PeerJ) and either DOI or URL of the article must be cited.
License URL: https://creativecommons.org/licenses/by/4.0/

Keywords: Conservation planning, Optimization, Prioritization, Integer linear programming, Prioritizr, Marxan

Funding: Liber Ero Fellowship and Environment and Climate Change Canada (ECCC) Cornell Lab of Ornithology Natural Sciences and Engineering Research Council of Canada and ECCC Richard Schuster is supported by a Liber Ero Fellowship and Environment and Climate Change Canada (ECCC), Jeffrey O Hanson by ECCC, Matthew Strimas-Mackey by endowments at the Cornell Lab of Ornithology, and Joseph R. Bennett by Natural Sciences and Engineering Research Council of Canada and ECCC. The funders had no role in study design, data collection and analysis, decision to publish, or preparation of the manuscript.

==============================
The resources available for conserving biodiversity are limited, and so protected areas need to be established in places that will achieve objectives for minimal cost. Two of the main algorithms for solving systematic conservation planning problems are Simulated Annealing (SA) and exact integer linear programing (EILP) solvers. Using a case study in BC, Canada, we compare the cost-effectiveness and processing times of SA used in Marxan versus EILP using both commercial and open-source algorithms. Plans for expanding protected area systems based on EILP algorithms were 12–30% cheaper than plans using SA, due to EILP’s ability to find optimal solutions as opposed to approximations. The best EILP solver we examined was on average 1,071 times faster than the SA algorithm tested. The performance advantages of EILP solvers were also observed when we aimed for spatially compact solutions by including a boundary penalty. One practical advantage of using EILP over SA is that the analysis does not require calibration, saving even more time. Given the performance of EILP solvers, they can be used to generate conservation plans in real-time during stakeholder meetings and can facilitate rapid sensitivity analysis, and contribute to a more transparent, inclusive, and defensible decision-making process.

Introduction

Area-based systematic conservation planning aims to provide a rigorous, repeatable, and structured approach for designing new protected areas that efficiently meet conservation objectives (Margules & Pressey, 2000). Historically, spatial conservation decision-making often evaluated parcels opportunistically as they became available for purchase, donation, or under threat (Pressey et al., 1993; Pressey & Bottrill, 2008). Although purchasing such areas may improve the status quo, such decisions may not substantially and cost-effectively enhance the long-term persistence of species or communities (Joppa & Pfaff, 2009; Venter et al., 2014). Systematic conservation planning, on the other hand, is a multi-step process that involves framing conservation planning problems as optimization problems with clearly defined objectives (e.g., minimize acquisition cost) and constraints (Margules & Pressey, 2000). These optimization problems are then solved to obtain candidate reserve designs (termed solutions), which are used to guide protected area acquisitions and land policy (Schwartz et al., 2018). Due to the systematic, evidence-based nature of these tools, they can help contribute to a transparent, inclusive, and more defensible decision-making process (Margules & Pressey, 2000).

Today, Marxan is the most widely used systematic conservation planning software, having been used in 184 countries to design marine and terrestrial reserve systems (Ball, Possingham & Watts, 2009). Although Marxan supports several algorithms for solving conservation planning problems, most conservation planning exercises use its implementation of simulated annealing (SA), an iterative, stochastic metaheuristic algorithm for approximating global optima of complex functions (Kirkpatrick, Gelatt & Vecchi, 1983). By conducting thousands of simulations to determine the impact of different candidate solutions, Marxan aims to generate solutions that are near-optimal. One of the reasons why Marxan uses SA instead of exact integer linear programing (EILP) solvers, is that EILP solvers were historically not well suited to solve problems with nonlinear constraints and penalties, such as problems trying to create spatially compact or connected solutions (i.e., compactness and connectivity goals) and generally took considerably longer than SA to solve problems (Sarkar et al., 2006; Haight & Snyder, 2009). However, the SA approach provides no guarantee on solution quality, and conservation scientists and practitioners have no way of knowing how close to optimal their solutions are. In this case, “optimal” refers to the configuration of protected areas that delivers the desired benefits and the lowest cost. The discussion about the relative merits of linear programing versus heuristics such as SA in conservation planning spans more than two decades (Cocks & Baird, 1989; Underhill, 1994; Church, Stoms & Davis, 1996; Rodrigues & Gaston, 2002; Önal, 2004), but the EILP shortcomings mentioned above have largely been overcome in recent years (Beyer et al., 2016).

In a recent simulation study, Beyer et al. (2016) found that Marxan with simulated annealing can deliver solutions that are orders of magnitude below optimality. They compared Marxan to EILP (Wolsey & Nemhauser, 1999), which minimizes or maximizes an objective function (a mathematical equation describing the relationship between actions and outcomes) subject to a set of constraints and conditional on the decision variables (the variables corresponding to the selection of actions to implement) being integers (Beyer et al., 2016). Unlike metaheuristic methods such as SA, prioritization using EILP will find the optimal solution or can be instructed to return solutions within a defined level of suboptimality. Some have argued that EILP algorithms are well-suited for solving conservation planning problems (Cocks & Baird, 1989; Underhill, 1994; Rodrigues & Gaston, 2002), but until recent advances in computational capacity and algorithms, it has been impossible to solve the Marxan-like systematic conservation planning problems with EILP for large problems (Haight & Snyder, 2009; Beyer et al., 2016).

Here we compare EILP solvers with simulated annealing as used in Marxan, for solving minimum set systematic conservation planning problems (Rodrigues, Cerdeira & Gaston, 2000) using real-world data from Western North America. The goal of solving the minimum set problem is to find the places that maximize biodiversity, while minimizing reserve cost. We found that EILP generated high quality solutions 1,000 times faster than simulated annealing that could save over $100 million (or 13%) for realistic conservation scenarios when compared to solutions obtained from simulated annealing. These results also hold true for problems aiming for spatially compact solutions. Our findings open up new possibilities for scenario generation to quickly explore and compare different conservation prioritization scenarios in real-time.

Materials and Methods

Study area

We focused on a 27,250 km2 portion of the Georgia Basin, Puget Trough and Willamette Valley of the Pacific Northwest region spanning the US and Canada, corresponding to the climate envelope indicative of the Coastal Douglas-fir (CDF) Biogeoclimatic zone in southwestern British Columbia (Meidinger & Pojar, 1991) (Fig. S1). Land cover in the region is diverse, with approximately 57% of the land in forest, 8% as savanna or grassland, 5% in cropland, 10% being urban or built and the rest in wetland, water or barren.

Biodiversity data

We used species distribution models for 72 bird species as our conservation features at a 1-ha grid cell resolution (Table S1). The distribution models were based on data from eBird, a citizen-science effort that has produced the largest and most rapidly growing biodiversity database in the world (Hochachka et al., 2012; Sullivan et al., 2014). From the 2013 eBird Reference Dataset (http://ebird.org/ebird/data/download) we used a total of 12,081 checklists in our study area, then filtered these checklists to retain only those from March to June to capture the breeding season, <1.5 h in duration, <5 km traveled, and a maximum of 10 visits to a given location to improve model fit. Sampling locations <100 m apart were collapsed to one location, yielding 5,470 checklists from 2,160 locations, visited from 1 to 10 times and 2.53 times on average. The R package unmarked (version 0.9-9; Fiske & Chandler, 2011) provided the framework for all species distribution models, which necessarily include two parts: occupancy and detection (MacKenzie et al., 2002). This form of distribution modeling, also known as occupancy modeling, uses the information from repeat visits to a site to infer estimates of detectability of a species as well as estimates of probability of occurrence. For further details on biodiversity data see Rodewald et al. (2019).

Property layer and land cost

We incorporated spatial heterogeneity in land cost (Ando et al., 1998; Polasky, Camm & Garber-Yonts, 2001; Ferraro, 2003; Naidoo et al., 2006) in our plans by using property data and 2012 land value assessments from the Integrated Cadastral Information Society of BC. This process resulted in 193,623 properties for BC which were subsequently used as planning units (Schuster, Martin & Arcese, 2014). Property data, including tax assessment land values from Washington State came from the University of Washington’s Washington State Parcel Database (https://depts.washington.edu/wagis/projects/parcels/; Version: StatewideParcels_v2012n_e9.2_r1.3; Date accessed: 2015/04/30), as well as San Juan County Parcel Data with separate signed user agreement. The combined property layer included 1.92 million polygons. Property data, including tax assessment land values from Oregon State had to be sourced from individual counties, which included Benton, Clackamas, Columbia, Douglas, Lane, Linn, Marion, Multnomah, Polk, Washington and Yamhill. The combined property layer for Oregon included 605,425 polygons. We converted the polygon cost values to 1-ha raster cells for consistency with the biodiversity data by calculating area weighted mean values of cost per raster cell. Using tax assessment values as an estimate of conservation cost is an underestimate because tax assessment values are often lower than market value, but estimates of market values over larger areas are rarely available and tax assessments do provide a good general approximation.

Spatial prioritization

We compared EILP and SA for solving the minimum set spatial prioritization problem (Ball, Possingham & Watts, 2009). In this formulation, the landscape is divided into a set of discrete planning units. Each planning unit is assigned a financial cost (here we use the assessed land value) and a conservation value for a set of features that we wish to protect (here the occupancy probability for a set of species). We also define representation targets for each species as the amount of habitat we hope to protect for that species. The goal of this prioritization problem is to optimize the trade-off between conservation benefit and financial cost (McIntosh et al., 2017). Achieving this goal involves finding the set of planning units that meets the conservation targets for the minimum possible cost (i.e., min cost: such that conservation value ≥ target). Details on the Marxan problem formulation can be found in Ball, Possingham & Watts (2009) and the EILP formulation in Beyer et al. (2016) and Appendix S2. Three key parameters that are important for Marxan analysis, which we also use here are: species penalty factor, number of iterations, and number of restarts (Ardron, Possingham & Klein, 2010). Briefly, the species penalty factor is the penalty given to a reserve system for not adequately representing a feature, the number of iterations determines how long the annealing algorithms will run, and the number of restarts determines how many different solutions Marxan will generate (for more details see Appendix S1). For all scenarios, we used 1 km2 planning units, generated by aggregating the species and cost data to this coarser resolution from the original 1-ha cells. Aggregation was accomplished by taking the sum of cost data and the mean of species data for all 1-ha cells within the larger 1 km2 cells.

EILP solvers (commercial vs open source)

A variety of EILP solvers currently exist, and both commercial and open source solvers are available. All solvers yield optimal solutions to EILP problems, but there are substantial differences in performance (i.e., time taken to solve a problem) and in the size of problems that can be solved (Lin et al., 2017). For the purposes of performance testing we opted for one of the best commercial solvers currently available, Gurobi (Gurobi Optimization Inc., 2017). In a recent benchmark study, Gurobi outperformed other solver packages for more complex formulations and a practical use-case (Luppold, Dominic & Heiko, 2018). To investigate solver performance of packages that are freely available to everyone, we also tested the open source solver SYMPHONY (Ralphs et al., 2019). Both Gurobi and SYMPHONY can be used from R. For Gurobi we used the R package provided with the software (Gurobi version 8.1-0) and for SYMPHONY the Rsymphony package (version 0.1-28; Harter et al., 2017). We used the prioritizr R package to solve EILP problems for both Gurobi and SYMPHONY solvers (Hanson et al., 2019).

Scenarios investigated

We investigated a range of scenarios that were computationally feasible for this study. For both Marxan and prioritzr we created the following range of scenarios: (i) vary conservation targets between 10 and 90% protection of features in 10% increments (nine variations), using (ii) 10–72 features (five variations) as targets, and (iii) with spatial extents of 9,282 planning units, 37,128 planning units, and 148,510 planning units (three variations), resulting in a total of 135 scenarios created (Table 1). For Marxan, we also varied two additional parameters, (i) the number of iterations ranged from 104 to 108 (five variations) and (ii) species penalty factors (SPF) of 1, 5, 25 and 125 were explored (four variations, roughly spanning two orders of magnitude) for a total of 2,700 scenarios investigated in Marxan (Table 1). Exploring ranges of values for number of iterations and SPF is recommended for calibration of Marxan to increase its ability to approximate the optimal solution (Ardron, Possingham & Klein, 2010). As the processing time for the most complex problem in Marxan (90% target, 72 features, 148,510 planning units, 108 iterations) was >8 h, we restricted the full range of scenarios to those mentioned above. The maximum number of planning units we used is within the range of previous studies using Marxan (Venter et al., 2014; Runge et al., 2016), although using more than 50,000 planning units with SA is discouraged without extensive parameter calibration, as near optimal solutions will be hard to find for problems of that size (Ardron, Possingham & Klein, 2010). To allow for a fair contrast between SA and EILP that focuses on algorithmic comparisons and not within SA variation, we focused our results and discussion on the best solution achieved with Marxan across 10 repeat runs.

Table 1 Scenarios investigated in our analysis.

The total number of scenarios tested for both Gurobi and SYMPHONY are 135. For Marxan analysis, we included calibration steps as well, which brought the total number of scenarios to 2,700 for that algorithm.

Parameter	Value range	Variations	Scenarios	
Targets	10–90%	9		
# Features	10, 26, 41, 56, 72	5		
# Planning units	9,282, 37,128, 148,510	3	135 (ILP)	
Marxan iterations	104, 105, 106, 107, 108	5		
Marxan SPF	1, 5, 25, 125	4	2,700 (SA)	

As systematic conservation planners often aim for spatially compact solutions to their problems, we also investigated a range of scenarios using a term called boundary length modified (BLM), which is used to improve the clustering and compactness of a solution (McDonnell et al., 2002). We randomly selected a 225 × 225 pixel region of the study area to generate a problem with 50, 625 planning units, the maximum recommended for Marxan. After initial calibration we set the number of features/species to 72, SPF to 25 and number of iterations for Marxan to 108. We varied targets between 10% and 90% protection of features in 10% increments, and used the following BLM values: 0.1; 1; 10; 100; 1,000 for a total of 45 scenarios. Both Marxan and prioritzr allow a user to specify BLM values as presented here. For details on the mathematical formulation of the spatial compactness constraint in ILP, please see Appendix S2 and Beyer et al. (2016).

All analyses were conducted on a desktop computer with an Intel Core i7-7820X Processor and 128 GB RAM running Ubuntu 18.04 and R v 3.5.3. All data, scripts and full results are available online (https://osf.io/my8pc/) and will be archived in a persistent repository with a DOI pending acceptance of the manuscript.

Results

Exact integer linear programming algorithms (Gurobi, SYMPHONY) outperformed SA (Marxan) in terms of their ability to find minimal cost solutions across all scenarios that met conservation targets. Summarizing across calibrated Marxan scenarios (number of iterations > 100,000 and species penalty factor 5 or 25), the range of savings ranged from 0.8% to 52.5% (median 12.6%, Fig. S2) when comparing EILP results to the best (cheapest) solution for a Marxan scenario. For example, at the 30% protection target EILP solvers resulted in solutions that were $55 million cheaper than SA (Fig. 1A), because the EILP solvers selected cheaper and fewer parcels in the optimal solution. With these savings an additional 961 ha could be protected (13,897 ha vs 12,936 ha) using an EILP algorithm by raising the representation targets until the cost of the resulting solution matched that of the Marxan solution using SA. In general, SA performed reasonably well at smaller problem sizes, fewer planning units and features and low targets, but as the problem size and complexity increased SA was less consistent in finding good solutions (Fig. S2). Cost profiles across targets, number of features and number of planning units are shown in Figs. S3–S5.

Figure 1 Solution cost and time comparisons.

(A) The lines represent costs compared to the Gurobi cost baseline. The numbers on the blue line represent total cost of a solution in million $ and the numbers on the green line represent how much more expensive, again in million $, the SA/Marxan solution is compared to the ILP solutions. (B) Time to solution comparisons between solvers. Marxan parameters used are: 72 features, 37,128 planning units, 107 iterations, using mean cost and time, across all Marxan runs that met their target for a given scenario (max = 10). Note that in (A) gurobi (red) and Rsymphony (blue) yielded optimal solutions for all target values and so their lines are plotted exactly on top of each other.

The shortest processing times were achieved using the prioritizr package and the commercial solver Gurobi, followed by prioritizr and the open source solver SYMPHONY, and lastly Marxan (Fig. 1B). Gurobi had the shortest processing times across all scenarios investigated, SYMPHONY tied with Gurobi in some scenarios and took up to 78 times longer than Gurobi in other scenarios (mean = 14 times, Fig. S6), and Marxan took between 1.8 and 1,995 times longer than Gurobi (mean = 281 times, Fig. S7). The longest processing times for Gurobi, SYMPHONY and Marxan for a single scenario were 40 s, 31 min and 8 h respectively. For the most complex problem (i.e., targets = 90%, 72 features; 148,510 planning units), Marxan calibration across the five number of iterations and four species penalty factor values took a total of 5 days 7 h, compared to 30 s using Gurobi and 28 min using SYMPHONY. Time profiles across targets, number of features and number of planning units are shown in Figs. S8–S10.

Exact integer linear programming algorithms (Gurobi, SYMPHONY) also outperformed SA (Marxan) when using a BLM to achieve more compact solutions. This was true for objective function values (Fig. 2A) as well as for processing times (Fig. 2B). Through finding optimal solutions, using EILP resulted in objective function values 5.65 to 149% (mean 22.7%) lower than SA values. Gurobi was the fastest solver to find solutions to problems including BLM in 44 of 45 scenarios, in one case SYMPHONY was faster. SYMPHONY outperformed Marxan in 44 of 45 scenarios, and took on average 13.7 times as long as Gurobi to find a solution (range −0.31 to 42.6). Marxan was never faster than Gurobi and took on average 104.6 times as long as Gurobi to find a solution (range 3.09–190.8). An example of the spatial representation of the solutions for a 10% target is shown in Fig. S11.

Figure 2 Objective function value and time comparisons using a boundary penalty to achieve spatially compact solutions.

(A) Deviation from lowest objective function value for solvers used and over a range of boundary penalty or boundary length modifier values (BLM); zero deviation indicates optimal solution. (B) Time to solution comparisons between solvers and across BLM values. Note that in (A) gurobi (red) and Rsymphony (blue) yielded optimal solutions for all target values and so their lines are plotted exactly on top of each other.

Discussion

We found that EILP algorithms outperformed SA both in terms of cost-effectiveness and processing times, even when including linearized non-linear problem formulations, when planning for spatially compact solutions. There have been calls for using EILP in solving conservation planning problems in the past (Underhill, 1994, Rodrigues & Gaston, 2002), but we are now at a point where making this switch is both advisable and computationally feasible, where technical capacity exists. Our study provides a systematic test, using real world data to build on the findings of (Beyer et al., 2016), and shows that their results hold for a realistic case study. We further expanded the scope of testing to include assessed land values in order to give estimates of how much better optimal solution can perform in terms of cost savings, compared to SA solutions. Finally, we showcase that even open source EILP solvers are much faster than SA algorithms as implemented in Marxan, which is very encouraging for non-academic user that would otherwise have to buy Gurobi licenses (Gurobi is free for academic use). The combination of the superior performance findings by both (Beyer et al., 2016) and this study indicates that EILP approaches should be strongly considered as improvements for minimum set conservation planning problems, currently solved using SA. This improvement is especially important in real world applications as the speed of generating solutions can be advantageous in iterative and dynamic planning processes that usually occur when planning for conservation (Sarkar et al., 2006). Given Marxan’s flexibility to use optimization methods other than SA, we hope that a future version of Marxan will include EILP solvers.

One practical advantage of using EILP over SA is that the analysis does not require parameter calibration. Unlike EILP, parameter calibration is a crucial task in every Marxan/SA project and the species penalty factors, number of SA iterations, and number of SA restarts must be calibrated to improve solution quality (Ardron, Possingham & Klein, 2010). This task can be very time consuming, especially for larger problems (e.g., 50,000 planning units). Ideally all possible combinations of parameters should be explored, but this further increases processing time. For instance, exploring three different parameter values would result in 27 different scenarios to explore (i.e., 3 × 3 × 3). Although we omitted calibration runs prior to finalizing and presenting results in this study, the parameter calibration step took several days for the most complex problem we investigated in this study. Yet none of this calibration time is necessary using EILP. An added benefit is that the somewhat subjective process of setting values for these three parameters can be eliminated using EILP as well.

Recommended practices for Marxan analyses caution against using SA for conservation planning exercises with more than 50,000 planning units (Ardron, Possingham & Klein, 2010). Such large-sized problems have occurred in the past and, as increasingly high resolution data become available, may become more common in the future (Venter et al., 2014; Runge et al., 2016). Unlike SA, EILP/prioritizr can solve problem sizes with more than one million planning units (Hanson, 2018; Schuster et al., 2019). Realistically, as problem sizes grow beyond what was intended for Marxan/SA projects, EILP will run into problems solving very large problems (>1 million planning units) that include non-linear constraints, such as optimizing compactness or connectivity, as those problem formulations need to be linearized for EILP to work. A potential future solution to this issue could be the use of nonlinear integer programing for more problems including non-linear constraints (Grossmann, 2002; Lee & Leyffer, 2011). Whether EILP would also outperform SA for more complex problem formulations, such as dynamic problems or problems with multiple objectives, still needs to be explored. Potential solutions would be to linearize the problem, or incorporate algorithms like Mixed Integer Quadratically Constrained Programming (Franco, Rider & Romero, 2014).

Finally, we argue that another strength of EILP solvers, especially Gurobi, is that they can be used to quickly explore and compare different conservation prioritization scenarios in real-time. This ability could be used to great advantage during stakeholder meetings, to explore various scenarios and undertake rapid sensitivity analysis.

Conclusion

Exact integer linear programming algorithms substantially outperform SA as used in minimum set systematic conservation planning, both in terms of solution cost, as well as in terms of time required to find near optimal or optimal solutions. Using an EILP algorithm, as implemented in the R package prioritizr, has the added benefit that users do not need to worry about or set parameters such as species penalty factors or number of iterations, which significantly reduces the time a user spends on finding suitable values for these parameters. Given the potential EILP is showing for conservation planning, we recommend users consider adding this modified approach to solving systematic conservation planning problems.

Supplemental Information

Supplemental Information 1 Details on important Marxan terminology, details on the integer programing formulation including equations, the list features used, a figure of the study area and additional results figures.

Click here for additional data file.

We thank W. Hochachka for providing code fore processing eBird data and three reviewers for insightful comments.

Additional Information and Declarations

Competing Interests

Author Contributions

Data Availability

The authors declare that they have no competing interests.

Richard Schuster conceived and designed the experiments, performed the experiments, analyzed the data, prepared figures and/or tables, authored or reviewed drafts of the paper, and approved the final draft.

Jeffrey O. Hanson conceived and designed the experiments, performed the experiments, analyzed the data, authored or reviewed drafts of the paper, and approved the final draft.

Matthew Strimas-Mackey conceived and designed the experiments, performed the experiments, analyzed the data, authored or reviewed drafts of the paper, and approved the final draft.

Joseph R. Bennett conceived and designed the experiments, authored or reviewed drafts of the paper, and approved the final draft.

The following information was supplied regarding data availability:

All data, scripts and full results are available at OSF: Schuster et al. (2019): “Exact integer linear programing solvers outperform simulated annealing for solving conservation planning problems”. OSF. dataset. https://osf.io/my8pc/.

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
