# Peer review of "Exact integer linear programming solvers outperform simulated annealing for solving conservation planning problems"

_PeerJ, doi:10.7717/peerj.9258_

## Round 0.1 · original submission · Minor Revisions

We have received three in-depth reviews of your paper, indicating that the paper is overall a valuable contribution, but that there are specific issues that need to be addressed in a revision. I look forward to reading a revised version soon.

Reviewer 1 ·

Basic reporting

Well presented manuscript with good placement in the literature

Experimental design

Clear experimental design and description of study.

Validity of the findings

conclusions well supported with data and follow logically from the description of the study design

Additional comments

This study explores the performance of a new approach to solving the spatial conservation prioritization problem that currently is commonly currently solved with the simulated annealing approach in Marxan. Recent advances in integer linear programming (ILP) have enabled the scale of problem that Marxan can handle by approximating optimal solution to now be feasible with an ILP approach, which can find an optimal solution. An R package (prioritizR) provides this functionality to conservation planners and researchers. Here, the authors explore the variation in recommended conservation portfolio (suggested protected areas) and the time to solution for a set of conservation scenarios that are run in Marxan, prioritizR with the optimization performed with Gurobi or prioritizR with the optimization performed with Rsymphony (two different options that differ in their availability to practitioners vs academics). They find that on all fronts, the Gurobi based solutions were superior (met the conservation targets at lower cost) and drastically faster than Marxan. The Rsymphony based solutions also were cheaper and faster than Marxan, although slower than Gurobi.

This paper is well written and generates important evidence supporting a new spatial conservation approach that can outperform the current industry leading approach. I have provided several suggestions for improvement and small edits but in general I think this paper is well presented, clearly written, and nicely executed.

Broad comments:
1. I would have liked to see a little bit more information in the introduction on what the advances in ILP were that now permit it to handle nonlinear constraints and penalties. I expect that this is covered in the Haight and Snyder, or Bayer references, but a sentence or two of the technical details would make a reader more convinced that the time is right for a move away from simulated annealing.
2. It was not clear whether the broad range of cost savings that could be achieved from the ILP over Marxan approach was due solely to the variation in problem specifications or if there was also an aspect of Marxan potentially providing different solutions to the same problem across different runs (because of the simulated annealing). Please add a bit of detail that comments on the variation due to difference in approach as opposed to the variation due to marxan’s approximate solutions.

Editorial comments.
Lines 22-23: It would help the reader if you could specify that this reduction in cost was due to being able to calculate the optimal solution as opposed to an approximation.
Line 53: Clarify that it ILP “historically” was not well suited
Line 59: It would be good to detail here what an optimal solution would be for readers who are not well versed in the terminology (e.g., something like “optimal being the configuration of protected areas that delivers the desired benefits and the lowest cost”).
Line 77-78: As a reader I wanted to know here how the ILP inability to deal with spatially compact solutions had been resolved
Line 100: I’m curious how you dealt with the repeated measures aspect of the sampling locations that have been visited multiple times. Is this something that unmarked has a strategy for accounting for?
Line 111-112: it might be worth mentioning that the use of tax assessment values as an estimate of conservation cost is an underestimate because tax assessment values are often lower than market value
Line 162: I initially read this as species per features and was confused. I think you actually mean species (e.g. features)
Line 162: the numbers run together and make it seem like one really large number.
Line 163-167: Please explain here why you wanted to explore additional Marxan configurations. I think it was to try configurations that increased Marxan’s ability to approximate the optimal solution but not sure from the text.
Lines 174-182: How did you specify the equivalent problem (BLM modifications) in ILP?
Line 195: Was the reduction because the ILP solution was able to find cheaper parcels or less parcels?
Figure 1: Are the results shown for Marxan from a single run? How much variation might we expect around this line if there were multiple Marxan runs? (I don’t think they are necessary, just a comment on the expected variability in Marxan solutions)

Reviewer 2 ·

Basic reporting

Well-written, professional throughout. All sufficient in this section.

Experimental design

All fine, although see comments on the framing of the research question in comments below.

Validity of the findings

All fine, although see major comments 1&2 below which will affect the framing/wording of the findings.

Additional comments

This manuscript makes an excellent point – Marxan is used as the gold-standard in conservation planning, but it is often not the best choice of algorithm. The manuscript is well-written overall, however, at the moment some components of the manuscript are lacking in this current form.

First, the authors have stayed away from a mathematical description of the problems being solved here. I think that is a mistake, and explicitly including the mathematical formulations could clear up or shine a light on a few of my major issues below (points 3-5). Second, some language throughout is misleading, which I detail in points 1 and 2 below. Other major issues follow, but I will point out that I think all of these are fixable with some very careful reframing/rewriting, and some different choices in the results section.

I do not think any of these issues call into question the fundamental relevance of this work to the conservation planning literature, and I hope the authors can clarify these issues in the manuscript since I hope to see it published.

Major issues

1. The dichotomy between ILP and SA presented here is incorrect, but I think it is just a terminology issue. Integer Linear Programs are a class of problem, with a specific canonical form. Reserve site selection problems largely fit this class. ILP itself is not a solution method/algorithm – in fact, SA is a valid (heuristic) solution method for ILPs. I think the comparison the authors are looking for here is not ‘SA vs ILP’, but ‘Solving ILPs with SA vs [an exact method -- branch and bound, CPLEX, or Gurobi etc]’.

2. Relatedly, the manuscript sometimes uses SA and Marxan interchangeably. Marxan uses a specific problem formulation in order to capture the intended problem (specifically by adding their Boundary Length Modifier), but then uses SA to solve that problem. It’s not clear until fairly late in this manuscript whether the authors are comparing:
• Simulated annealing vs an exact optimisation method to solve the same problem (same objective function, identical constraints); or
• The Marxan formulation of the reserve site selection problem vs an ILP formulation of the reserve site selection problem solved using an exact optimisation method.
The first half of the manuscript feels like it is setting up the first comparison, but the latter is what ends up being done – but it is not clearly stated, the reader is left to figure it out. This manuscript compares two ILP solvers (Gurobi and SYMPHONY) with Marxan. Marxan is a specific implementation of a simulated annealing algorithm, and is built from a very specific problem formulation. I do not think it is fair to claim this is a comparison of ILP solvers and simulated annealing in general for conservation planning problems, it is a comparison between ILP solvers and Marxan.

3. In Materials and Methods, it would be very useful to introduce the problem being solved in either the first or second section. The last parag of the introduction mentions a ‘systematic planning problem’, but it is not yet clear what this actually means in this context.

4. The Spatial Prioritization section would be easier to follow with a mathematical description of the ILP being tackled (i.e. write out the min set problem, with all constraints), and the Marxan problem. This is related to my issues with the dichotomy elsewhere – this is where that could be cleared up by embracing the math!

5. Is there an analogue to the BLM in the ILP formulation used? How does the ILP formulation take into account clumpiness? One of the big arguments for Marxan is the ability to control or explore how clumped the solutions are, so that needs to be addressed here.

6. Much of the M&M section is written assuming a fairly good knowledge of Marxan. There are some terminology that need to be cleared up:

Species Penalty Functions
Boundary Length Modifiers
Calibration
Iteration
Maybe more that I have missed since I know Marxan, give it a good scour for other jargon.

7. In Fig 2, the two objective functions are different for the two problem formulations. Is this deviation in the actual lowest objective function value (which includes a BLM penalty), or the cost of the optimal solution? The latter would be a fairer comparison. For example, the BLM of 100 performs much worse at a 60% target, but it is not clear to me whether that’s just because a larger penalty is being applied to the objective function.

8. The two results figures use the 10^8 iterations version of the Marxan runs. That seems like a lot – is that standard? It may be a fairer comparison to examine the times at the median number of iterations tested, which would drop the times back a lot. Figure 1 could actually be cool with multiple lines for Marxan, showing multiple interation choices – I think with only a few, it wouldn’t be distracting.

9. The same point can be made for the number of planning units. The discussion notes that the standard recommendation is not to use Marxan for more than 50,000, yet the results presented here are for more than three times that size. Do the general observations here hold true for a problem where Marxan would actually be considered useful? I’d like to see the results and the figures for a more realistic problem more representative of how Marxan is used in practice. If the message here is really that these other approaches can solve much larger problems than Marxan, then the messaging needs to be changed throughout the manuscript.

10. Although I agree that this is strictly a conservation planning paper (which is fine), it would be good to look outside the field for similar discussions and analyses. The tradeoffs between SA and ILP, or even heuristic vs exact optimisation methods is not a brand new one. The comparison is certainly novel here in a field that so heavily relies upon SA, but a richer context could be drawn from other literatures.


Minor issues

Table 1 is not referred to in the text where it would be very helpful, and it is hard to parse.

Line 51 ‘performing runs’ is a little jargony for this early in an introduction – would be good instead to make it clearer at this point that SA is expensive because it performs many hundreds of simulations to determine the impact of different candidate solutions.

Line 53 – talking about the drawbacks (e.g. the structure of the problem and the time to solve) of ILP in past tense, can that be contextualised that more? Have these problems been solved and are no longer issues, or are you proposing that they are worth the tradeoff?

Line 58 – ‘highly suboptimal’ is confusing wording

I do not know what cadastral data means, sorry.

Line 110 mentions polygons – the context for these has not been set yet. I suggest adding this to the section I suggested at the beginning of M&M describing the problem.

Line 126 – is land cost a socioeconomic cost? I’d be more inclined to call this a financial cost, since there isn’t anything social being captured.

Line 191 – it is disingenuous to present results from uncalibrated Marxan runs, especially when it results in something so extreme as a cost saving of over four thousand percent. That would be a very powerful number if it came from the intended use of Marxan!

Line 197 I really like the comparison as illustrated here – how much more you could select in the ILP formulation if you had the same budget as the Marxan optimal solution. If there are other obvious places to make this comparison in the results, I’d love to see this illustrated again. It may not fit neatly anywhere else though, which is fine.

·

Basic reporting

All good.

Experimental design

All good.

Validity of the findings

All good. I note some of the interpretations go beyond what the paper shows evidence of and have made some suggestions below.

Additional comments

Thanks for the opportunity to review this paper. Sorry for my delay.

The authors conduct a real- world analysis to support already established findings that ILP is faster and more efficient than Simulated Annealing when it comes to solving the minimum set problem- the most common problem definition in spatial conservation prioritization.

From a technical perspective, the authors present their case clearly. The analysis is robust and the findings do not deviate from previous examples where similar comparisons have been made (e.g. Beyer et al. 2016). To have a real-world example, rather than a simulation-based analysis, is useful for users and a nice contribution to the conversation about tools to underpin planning processes. As a technical document, I think this work is helpful, and having a discussion around open source vs proprietary ILP solvers will be useful for many people looking to use ILP in their analyses.

Where I do have some concerns is in the framing towards broader conservation planning guidance and recommendations. While the authors use real-world data, I do not see clear real-world planning evidence offered in the paper to support some of these broader claims.

For example, Line 39: I would disagree that systematic conservation planning is about framing optimization problems. SCP is a ten step process, only one step is about framing and solving the problem for prioritization, the rest is about dialogue, communication, stakeholders, transparency, structure and policies. A SCP process can take 10 years- running a tool takes a week. We should be careful to not over-emphasise the role of tools and algorithms in the much larger effort of SCP. In the end, the algorithm is only a supporting application and the same can be said for the other planning softwares like Marxan and Zonation.

Lines 232: the authors encourage “making the switch to ILP as being advisable now that it is computationally feasible” to support SCP processes. I think adding in a clause about “where technical capacity exists” would be prudent. Many decision-makers and technicians in the places where we desperately need to deploy SCP approaches and tools are not even computer or spatially literate, and so I am not sure “who” this message is for and “who” is advising this switch. Some clarification and evidence is needed if a grand statement such as this is going to be made. I think this paper will be of great interest for technically savvy people working on large optimization problems where optimality and speed are desirable- but this is a different target audience than mid-level practitioners in Tanzania or Solomon Islands working to support government decision-making through SCP.

While reading this paper- the following analogy came to mind:
I find this paper analogous to being told to choose between a Porsche (ILP) and a pick-up truck (SA), and then being told the Porsche is what I should buy because on all counts, it performs much more efficiently than the pick-up truck. It will get me where I need to go faster than anything else. But what if my objective is to build a house? Would I still be told to choose the Porsche? According to the logic of this paper, the answer is yes, because speed and efficiency are the only things driving the choice. In reality, there will be a suite of external factors influencing the choice of car that go far beyond performance efficiency. The “one is superior to the other” message that continuously emerges in the paper, and which is solely driven by the technical efficiency of the algorithm, feels a bit myopic when the authors move beyond the algorithmic comparison to discuss its applicability for integration in broader planning processes.

Prioritzr, Zonation, and Marxan all aim to serve the same role: to support and improve transparency in decision-making for planning. They all have pros and cons and can be deployed in different ways for different users in different political and planning contexts. I would encourage the authors to remove statements suggesting the world transition to ILP because of its superior performance, if their only evidence for this claim is efficiency. To demonstrate otherwise would require a different comparison that is beyond the scope of this paper.

However, the ILP-based Prioritzr is a promising new addition to the suite of tools we have at our disposal for planning and I think with a minor tweak to some of the framing, this paper will be well-received amongst those individuals looking for more powerful solvers.

It might also be useful to suggest how ILP and SA can compliment each other in broader planning processes if the authors do want to bring the discussion up to a higher level than technical comparisons.

---

## Round 0.2 · accepted · Accept

The three reviewers and I agree that your revised manuscript has addressed any remaining issues and we believe it is a much improved presentation of the work.

Reviewer 1 ·

Basic reporting

Excellent

Experimental design

Excellent

Validity of the findings

Excellent

Additional comments

I have read through the revised version and the author's response and am happy with how they have addressed my and the other reviewer's comments. I have no additional comments to add and hope to see this paper published soon.

Reviewer 2 ·

Basic reporting

Good

Experimental design

Good

Validity of the findings

Good

Additional comments

Thanks for the opportunity to review this manuscript again. While some issues have been addressed well, some concerns remain after being insufficiently addressed in the revision and response document.


1. Remaining issues with the language:
Throughout, there are still places where ‘SA’ is referred to, but the authors specifically mean Marxan (i.e. it is not a generalizable statement about Simulated Annealing):
Line 276
Line 280
Line 281
Line 312
Line 321
Line 330
Line 387
Throughout, “EILP” is used to mean ‘Exact Integer Linear Programming’, but this should be followed by the word ‘solver’ in most cases. The ‘exact’ thing is the solver.

Likewise, in some places EILP is referenced but I believe the authors are referencing a specific implementation of ILP (probably prioritizr, but not clear):
Like 364, 372, 377

2. The Introduction is still unclear on why EILP is suddenly useful. It’s alluded to in the tail end of two paragraphs, but what are the changes that mean EILP has overcome all the issues it had that allowed the meteoric rise of SA? Without this motivation this paper reads as if the authors have just heard of EILP and thought “why not have a crack” – which would be fine, except it implies that the use of SA over the years has somehow been blind to the existence of EILP. This is a naive implication, and as currently written the paper seems so evangelically in favour of EILP that its drawbacks cannot be touched on, which makes the reader (maybe incorrectly) suspicious of the findings. The addition of the sentence at line 76 since the previous review is insufficient to mitigate the appearance of this attitude.


3. Methods section has no description of the problem – what are you solving? This relates to my point in the first review:
‘4. The Spatial Prioritization section would be easier to follow with a mathematical description of the ILP being tackled (i.e. write out the min set problem, with all constraints), and the Marxan problem. This is related to my issues with the dichotomy elsewhere – this is where that could be cleared up by embracing the math!’
In my opinion, adding a section to the Supplementary Information is not sufficient – it is not clear from reading the Methods section what that objectives or constraints of the problem are. The readership is limited to those who deeply understand the underlying Marxan and reserve site selection formulations.

4. The lacking objective function definition also makes the response to my comment about Figure 2 insufficient – the y axis is being measured by a function that is not in the main text of the paper, and therefore is very opaque.


Minor: Line 104 ‘minimum set’ has not been described yet. Use a different, less jargon-laden description.

·

Basic reporting

The revision is a much improved version of the manuscript.

Experimental design

The scenarios are well developed and the testing is rigorous and comprehensive.

Validity of the findings

The findings reconfirm what is already known. The results are presented clearly and make sense. The license versus open source EILP comparison remains a nice, new addition to the topic of discussion.

Additional comments

I think the previous review process was very helpful. Particularly the attention paid to differentiating ILP as a class of problem and the EILP solvers used in the analysis.

I think the paper is acceptable to publish and have no further comments.